# ON TRANSFERRING EXPERT KNOWLEDGE FROM TABULAR DATA TO IMAGES

## ABSTRACT

Transferring knowledge across modalities has gained considerable attention in machine learning. Expert knowledge in fields like medicine is often represented in tabular form, and transferring this information can enhance the accuracy of image-based learning. Unlike general knowledge reuse scenarios, tabular data is divided into numerical and categorical variables, with each column having a unique semantic meaning. In addition, not all columns in tabular can be accurately represented in images, making it challenging to determine "how to reuse" and "which subset to reuse". To address this, we propose a novel method called CHannel tAbulaR alignment with optiMal tranSport (CHARMS) that automatically and effectively transfers relevant tabular knowledge. Specifically, by maximizing the mutual information between a group of channels and tabular features, our method modifies the visual embedding and captures the semantics of tabular knowledge. The alignment between channels and attributes helps select the subset of tabular data which contains knowledge to images. Experimental results demonstrate that CHARMS effectively reuses tabular knowledge to improve the performance and interpretability of visual classifiers.

## 1 INTRODUCTION

Data takes on various forms, such as images, text, video, and audio, providing rich and diverse sources of information for a given task. In contrast to using a single modality, multimodal learning aims to fuse information from different data modalities to create more comprehensive and accurate models (Baltrušaitis et al., 2018; Ngiam et al., 2011; Ramachandram & Taylor, 2017; Yang et al., 2020). This approach has demonstrated exceptional performance across many domains, including recommender systems (Salah et al., 2020; Huang et al., 2019; Baltescu et al., 2022), healthcare (Zhang et al., 2022; Han et al., 2022), and visual question answering (Li et al., 2019; Zheng et al., 2020; Jing et al., 2020).

In practical applications, obtaining data from multiple modalities can be challenging (Zhou, 2018), as expert knowledge or specialized equipment may be required, such as medical images. The high acquisition cost of such data makes the traditional multimodal fusion approach impractical. To address this, one solution is to employ multiple modalities during training, enabling expert knowledge to transfer from one modality to another and improving the performance of a single modality during testing. The current research on crossmodal transfer primarily focuses on images and text (Karpathy & Fei-Fei, 2015; Wang et al., 2016; Radford et al., 2021), but limited exploration has been done with tabular data (Hager et al., 2023).

Tabular data is a common type of structured data, usually organized in a table format, where each column represents an attribute or feature and each row represents a sample of data (McKinney et al., 2010). Tabular data often involves some expert knowledge, for example, in the medical field, an attribute of tabular data may represent position information in an MRI image that needs to be focused on, which requires expert annotation. Therefore, transferring expert knowledge from tables to images will improve detection efficiency and reduce the burden on doctors. However, tabular data's structured format distinguishes it from existing unstructured data such as text, making existing crossmodal transfer methods unsuitable for tabular data (Kimball & Ross, 2011; Shwartz-Ziv & Armon, 2022).

Specifically, we face two challenges in transferring tabular knowledge for images. Firstly, we must address "how to reuse" the tabular data. As each column in tabular data has a unique semantic meaning, relying on standard RNN (Hopfield, 1982; Zaremba et al., 2014) or Transformer (Vaswani et al., 2017) methods to construct a coarse feature space would result in a loss of interpretability of certain attributes. Moreover, categorical and numerical variables in tabular data require different processing methods. Secondly, we must identify "what subset to reuse" from the vast amount of information contained in tabular data since not all of it is relevant to the corresponding image. For example, in a pet adoption scenario, the tabular data contains not only the type of the pet but also information such as whether the pet is vaccinated or not. Therefore it is crucial to identify the useful information that can be transferred to instruct the learning of images. We expect that by transferring tabular knowledge to an image model, the model can learn corresponding semantics more effectively and achieve better performance on correlation tasks.

To overcome the aforementioned challenges, we propose a novel method named CHannel tAbulaR alignment with optiMal tranSport (CHARMS) that aligns tabular data attributes with image channels which automatically transfers relevant expert knowledge in tabular data to images. Specifically, we modify the visual embedding with the instruction of tabular data as auxiliary information and learning tabular features with a group of channels, maximizing the mutual information between them. Additionally, we utilize the optimal transport algorithm (Bonneel et al., 2011; Caffarelli & McCann, 2010) to match the representation of each channel with the representation of each attribute, where a distinction is made between categorical and numerical variables. We strengthen the corresponding channels to ensure a focused learning of the tabular knowledge. In this way, our approach can automatically and effectively utilize expert knowledge from tabular data in the learning process, outperforming previous methods. To summarize, our contribution is three-fold:

- We emphasize the importance of knowledge transfer from tabular data to image data, as this can lead to improved performance when tabular data is missing due to high costs.
- We propose CHARMS method to automatically transfers relevant tabular knowledge to images. It aligns attributes and channels by leveraging optimal transport and utilizes tabular data as auxiliary information during transfer.
- Experimental results demonstrate that CHARMS effectively reuses tabular knowledge to improve the performance of visual classifiers. Moreover, our approach offers insightful explanations of the learned visual embedding space with tabular instruction.

This paper is organized as follows: the related work is introduced in Section 2. Section 3 and Section 4 provide the setting formalization, discovery experiment and our method. In Section 5, we present experiment results and discuss our findings. Finally, Section 6 concludes our study results.

## 2 RELATED WORK

**Multimodal Learning.** Data of different modalities, such as image, video, audio, and text, usually overlap in some content, while some information is complementary. Multimodal learning aims to leverage the information in different modalities to learn a better representation and improve the performance of the task for different scenarios. An important task in multimodal learning is the fusion of modalities. Some previous work used BERT (Li et al., 2020a; Su et al., 2019) or co-attention (Li et al., 2019; Tan & Bansal, 2019) to fuse different modal information simply. Subsequently, some large models (Li et al., 2021; Jia et al., 2021; Li et al., 2022) were created to align the information of different modalities in terms of their semantic relationships using contrastive learning approach (Tsai et al., 2018). Different pre-training approaches have also been extensively studied (Bao et al., 2022; Huang et al., 2021; Yao et al., 2021; Liang et al., 2020).

**Crossmodal Transfer.** The modality fusion approach directly depends on the integrity of the data from different modalities. However, the reality is often that we do not have access to the data of all modalities. Therefore, another direction of multimodal learning is to construct robust models to cope with missing modalities or crossmodal transfer. For example, knowledge in missing modalities can be complemented using autoencoders or generative adversarial approaches (Cai et al., 2018; Pan et al., 2021; Li et al., 2020b). Ma et al. (Ma et al., 2021) improves the robustness of Transformer models by automatically searching for an optimal fusion strategy regarding input data. Wang et al. (Wang et al., 2020) proposed a framework based on knowledge distillation, utilizing the supplementary information from all modalities, and avoiding imputation and noise associated with it.

Hager et al. (Hager et al., 2023) proposes the first self-supervised contrastive learning framework that takes advantage of images and tabular data to train unimodal encoders. But most of these approaches consider Vision-Language scenarios, audio or video, which have been well investigated and are not suitable for tabular data due to their structured character and the difference between numerical and categorical variables. Our approach fills the gap of multimodal learning on tabular modality by taking it into account.

**Learning with Tabular Data.** The learning of tabular data has become an important research direction in the field of machine learning and data science for a long time. Traditional machine learning methods have been widely used on some tabular data, such as decision trees (Quinlan, 1986), support vector machines (Vapnik, 1999), and random forests (Breiman, 2001). These methods usually rely on pre-processing steps such as manual feature engineering and data cleaning, followed by model training and prediction using supervised learning. With the development of deep learning, tabular modeling approach using deep learning (Wang & Sun, 2022; Huang et al., 2020; Gorishniy et al., 2021) is very appealing because this allows tabular data to be used as input to a single modality and trained end-to-end by gradient optimization, which is competitive with GDBT methods (Chen & Guestrin, 2016; Ke et al., 2017; Prokhorenkova et al., 2018). In recent years, more and more approaches for tabular data have been proposed (Arik & Pfister, 2021; Hollmann et al., 2022; Yan et al., 2023; Jeffares et al., 2023). However, tabular data usually contains expert knowledge, such as medical diagnosis information of doctors and seismic waveform information, making it costly to acquire. So we consider such a scenario. Expert knowledge from the tabular data is used to guide the learning of the image data during training, with the expectation that good performance can be efficiently obtained even when the tabular data is missing during testing.

## 3 PRELIMINARIES

In this section, we first introduce the crossmodal transfer task, followed by some existing methods and some analysis.

### 3.1 TRANSFER KNOWLEDGE FROM TABLE TO IMAGES

Formally, we define the crossmodal transfer training dataset $D_{train} = \{\boldsymbol{x}_i^T, \boldsymbol{x}_i^I, y_i\}_{i=1}^N$, where $\boldsymbol{x}^I \in \mathbb{R}^{H_0 \times W_0 \times C_0}$ represent image data, $\boldsymbol{x}^T \in \mathbb{R}^D$ represent tabular data and $y \in Y$ is the label space of the task. The image data is represented as a three-dimensional tensor with height $H_0$, width $W_0$, and RGB channels $C_0 = 3$, while the tabular data is a vector of dimension $D$, where each dimension corresponds to an attribute. We define the test dataset $D_{test} = \{\boldsymbol{x}_i^I\}_{i=1}^M$, where tabular modality is missing due to high collection cost and the need for expert annotation. During training, we aim to minimize the empirical risk of model $f(\boldsymbol{x})$ over the training set:

$$\sum\nolimits_{(\boldsymbol{x}_i^I, \boldsymbol{x}_i^T, y_i) \in D_{train}} \mathcal{L}(f(\boldsymbol{x}_i^I), y_i \mid \boldsymbol{x}_i^T), \tag{1}$$

where $\mathcal{L}$ is the loss function that measures the discrepancy between prediction and ground-truth label such as cross-entropy loss and $\mid$ indicates conditioning on the tabular data. The model can be decomposed into embedding and linear classifier: $f(\boldsymbol{x}) = \boldsymbol{W}^\top \phi(\boldsymbol{x})$, where $\phi(\cdot) : \mathbb{R}^D \to \mathbb{R}^d$ is the feature extractor to extract the embedding of the images and $\boldsymbol{W} \in \mathbb{R}^{d \times |Y|}$.

Our objective is to transfer relevant tabular information into the image model $f$. In situations where expert knowledge is not available, we expect the model to provide better predictions when only given the image data $\boldsymbol{x}^I$ on the test set.

### 3.2 METHODS FOR CROSSMODAL TRANSFER

One of the main challenges in this task is how to transfer the tabular knowledge to the image model. It is feasible to align the two modality and then select the appropriate part for knowledge transfer. So we explore methods with alignment from different perspectives, including output-based transfer, parameter-based transfer, and embedding-based transfer.

**Output-based Transfer.** To transfer knowledge from tabular data to image models, we aim to ensure that the predictions of image model $f$ and tabular model $g$ are aligned. To achieve this, we first train a classifier $g$ on the tabular data such as LightGBM (Ke et al., 2017). We then fit the prediction

results of the image model $f$ to $g$ during the training. Knowledge Distillation (KD) (Hinton et al., 2015) is an output-based method:

$$\mathcal{L}(\boldsymbol{x}^I, \boldsymbol{x}^T, y) = (1 - \lambda)\mathcal{L}(\boldsymbol{x}^I, y) + \lambda\mathcal{L}_{\text{KD}}(f(\boldsymbol{x}^I), g(\boldsymbol{x}^T)). \tag{2}$$

$\mathcal{L}_{\text{KD}}$ measures the similarity between the prediction of two models with Kullback-Leibler (KL) divergence $g$ is called teacher network and $f$ is student network. Aligning the output of the tabular model and the current model helps to reuse the knowledge in tabular data.

So as Modality Focus Hypothesis (MFH) (Xue et al., 2022), the modality general decisive information is set according to the feature importance (Breiman, 2001; Wojtas & Chen, 2020) in tabular data as the teacher network, selecting subset of the tabular data. Then only use $\mathcal{L}_{\text{KD}}$ for distillation to fully observe the tabular's influence on image.

**Parameter-based Transfer.** The parameters of the model may contain part of the knowledge in the data, so the knowledge can be transferred from the perspective of the parameters of the model as well. For example, Fixed Model Reuse (FMR) (Yang et al., 2017) utilizes the learning power of deep models to implicitly grab the useful discriminative information from fixed models/features. In our setting, the fixed features referred to here are the tabular data:

$$\mathcal{L} = y \log h\left(f\left(\boldsymbol{x}^I\right) + g\left(\boldsymbol{x}^T\right)\right) + \frac{1}{2}\left\|\boldsymbol{x}^T - \phi(\boldsymbol{x}^I)\boldsymbol{U}\right\|_F^2. \tag{3}$$

$h$ is a soft-max operator and $\boldsymbol{U}$ is the linear connections between the tabular features and embedding of images. To transfer the influence of the fixed features $\boldsymbol{x}^T$ to images during the training procedure, FMR removes those connected parts corresponding to features $\boldsymbol{x}^T$ gradually and finally vanish all related components with the knockdown method.

**Embedding-based Transfer.** The method expects to find a subspace in which the embedding of similar images and tabular data is as close as possible, while the embedding of dissimilar images is as far as possible. For example, Multimodal Contrastive Learning (MMCL) (Hager et al., 2023) proposes the self-supervised contrastive learning framework that takes advantage of images and tabular data to train unimodal encoders:

$$\mathcal{L} = \lambda\ell_{I,T} + (1 - \lambda)\ell_{T,I}, \quad z_{j_I} = f_{\phi_I}\left(\phi(\boldsymbol{x}^I)\right),$$
$$\ell_{I,T} = -\sum_{j \in \mathcal{N}} \log \frac{\exp\left(\cos\left(z_{j_I}, z_{j_T}\right)/\tau\right)}{\sum_{k \in \mathcal{N}, k \neq j} \exp\left(\cos\left(z_{j_I}, z_{k_T}\right)/\tau\right)}, \tag{4}$$

where embeddings are propagated through separate projection heads $f_{\phi_I}$ and $f_{\phi_T}$ and brought into a shared latent space as projections $x_{j_I}, z_{j_T}$. $\ell_{I,T}$ is calculated analagously. $\mathcal{N}$ denotes all subjects in a batch. Then MMCL uses linear probing of frozen networks to evaluate the quality of the learned representations. By mapping tabular and image data to the same space and utilizing contrastive learning methods, the knowledge in tabular data can be transferred into an image feature extractor.

While the output-based, parameter-based, and embedding-based methods offer perspectives on transferring knowledge between modalities, each method has its own limitations. The output-based approach offers a simple and straightforward alignment based on the output of the model, but it may not capture detailed information for a certain attribute. The MFH method considers important features, but it completely discards other information during knowledge distillation. Parameter-based methods such as FMR cannot address the significant differences between tabular and image models, and the information contained in the parameters may be limited. The embedding-based approach attempts to find a common subspace for alignment but may lose some attribute information in the tabular data when changing the space, potentially ignoring valuable expert knowledge during transfer. By exploring these different transfer methods and their respective limitations, we can gain a deeper understanding of the challenges and opportunities in multimodal learning and develop more effective approaches for transferring knowledge from table to images.

## 4  TRANSFERRING KNOWLEDGE AFTER ALIGNMENT

Motivated by the unique characteristics of tabular data, we leverage it as auxiliary information in our approach to transfer knowledge to the image modality. Specifically, we minimize the mutual information between the image and each attribute of the table data, effectively transferring the relevant table knowledge to the image modality. Additionally, we use Optimal Transport to match the

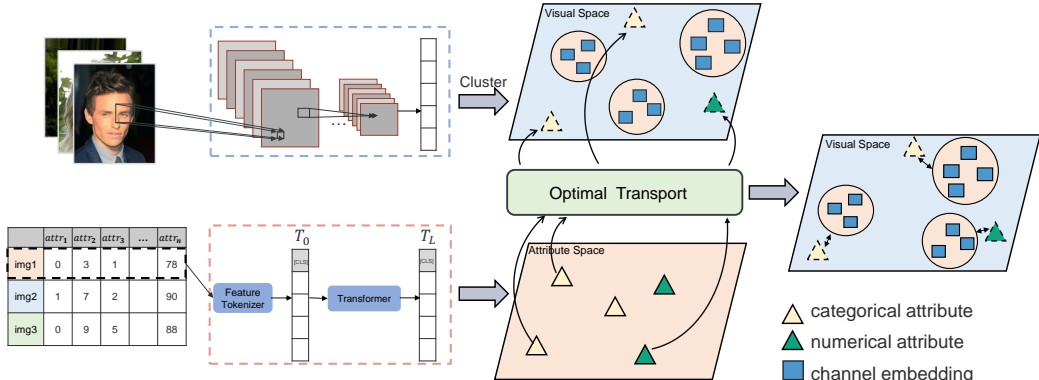

Figure 1: Flowchart of CHARMS method. Our approach combines the learning of image and tabular data, leveraging the specific characteristics of each modality to effectively transfer knowledge from one to the other. We use Optimal Transport (OT) methods to match tabular attributes to image channels, effectively learning the correlation attribute of the tabular data with the focused channels as a means of transferring expert knowledge to the images and solving the crossmodal transfer problem.

expert knowledge that can be expressed in the image data, allowing us to select a subset of the image features and strengthen the learning of the corresponding channels. Our approach highlights the importance of leveraging the specific characteristics of each modality to develop effective transfer. The flowchart is shown in Figure 1.

## 4.1 PRELIMINARY EXPERIMENTS

We evaluate the quality of crossmodal transfer with MINE method, which uses mutual information, a measure of information in information theory that quantifies the amount of information contained in one variable about another (Belghazi et al., 2018). In our setting, a good image model based on tabular knowledge transfer should contain more tabular knowledge, resulting in higher mutual information both with the image and tabular data. To evaluate our approach, we conduct experiments on MFEAT dataset (van Breukelen et al., 1998), using two types of tabular data: 76 Fourier coefficients of character shapes and 6 morphological features. The image modality is reconstructed from 240 pixel averages of images from $2 \times 3$ windows. The result is shown in Figure 2. The Tab-Only and Img-Only methods are the result of models trained on a single modality.

Our experiments indicate that existing methods for transferring tabular knowledge to image models yield low mutual information between the representations and tabular data. This suggests that these methods are not effective at transferring all types of tabular knowledge to the image modality and that feature selection is crucial. To validate this hypothesis, we perform knowledge distillation of the image model using two models trained on different parts of the tabular data. We find that morphological features in the tabular data can effectively promote image information, while other non-morphological features can make the tabular information more comprehensive.

These results highlight the importance of the careful selection of different tabular attributes and their relationship with the image modality. Similarly, different channels exist for the images, and the choice of different channels can

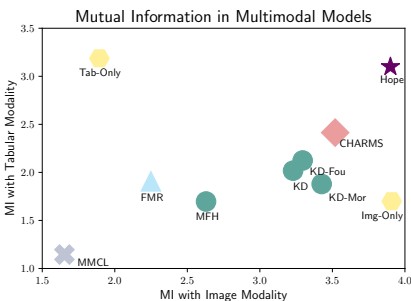

Figure 2: Mutual Information with Different Modality in Multimodal Models. A good crossmodal transfer model should be able to effectively combine both image and tabular information, resulting in higher mutual information between the two modalities. Ideally, the model should be positioned in the upper right corner.

also impact the final performance of the model. Since these methods do not transfer table information well, it is important to know how to use tabular knowledge. Based on these findings, we propose our method for transferring knowledge between modalities, which takes into account the specific characteristics of each modality and transfers expert knowledge to guide the image model.

## 4.2 Channel Tabular Alignment

To extract the relevant information from the tabular data that is beneficial to the image model, we also use alignment-based methods for feature selection. This task consists of two main parts: first, obtaining the intermediate embedding of the image and tabular data; and second, performing alignment-based feature selection.

To extract representations of the different channels, we use convolutional neural networks (CNNs). CNNs leverage convolutional filters to scan over the input data and extract local features. By stacking multiple convolutional layers, CNNs can learn increasingly complex and abstract features, allowing us to obtain different channels that capture different aspects of the image. Specifically, the channels of image data $x^I$ are defined as $\phi_{-1}(x^I) \in \mathbb{R}^{H \times W \times C}$, where $C$ is the number of channels, and each channel corresponds to a high-level feature such as edges, whose shape is $H \times W$.

Similarly, we use a neural network to obtain the representation of each attribute of the tabular data. This involves transforming all features, including both categorical and numerical variables, into embeddings. The resulting attributes are defined as $\psi(x^T) \in \mathbb{R}^{D \times E}$, where $D$ is the number of attributes and $E$ is the embedding dimension. We assume that the first $p$ attributes are numerical variables $x_{num}^T$, and the remaining $q$ attributes are categorical variables $x_{cat}^T$.

Secondly, we use the optimal transport to align the channels of the image with the attributes of the tabular data (Benamou et al., 2015). OT is a mathematical framework for measuring the similarity between probability distributions and finding the optimal way to transport mass from one distribution to another. The basic idea behind OT is to find a mapping between the elements of two distributions that minimizes the cost of moving one distribution to the other. The cost is typically defined as a distance metric between the elements. However, not all tabular attributes can be displayed on the image, and in some cases, there may be missing or irrelevant attributes that cannot be aligned with the image data. For example, on the PetFinder-adoption dataset, the photo of the pet can reflect the pet's hair, body size, and other attributes, but not the health condition or vaccination status. To address this issue, we use the partial optimal transport (POT) algorithm (Chapel et al., 2020).

Specifically, To address the issue that different channels of an image may have repeated semantics with some redundancy, we use K-Means (Lloyd, 1982; MacQueen, 1967) clustering to group similar channels together. This allows us to obtain fewer distinct channels, each capturing a distinct aspect of the image data. Then we compute the cosine similarity of the dataset on each channel, resulting in a matrix $S_I \in \mathbb{R}^{C' \times N \times N}$, where $C'$ is the number of clustered channels and $N$ is the length of the dataset. In parallel, we process the attributes of the tabular data similarly to obtain the attribute-wise similarity matrix $S_T \in \mathbb{R}^{D \times N \times N}$. Then the cost matrix is constructed from the channel-wise similarity between attribute-wise similarity. Then the OT transfer matrix is calculated:

$$C_{ij} = \left\| S_{T_i} - S_{I_j} \right\|_2^2, \quad T = \arg\min_{T} \langle C, T \rangle_F, \tag{5}$$

where $\langle \cdot \rangle_F$ denotes the Frobenius norm. After aligning the distributions of the image and tabular data using optimal transport, we obtain the transfer matrix $T \in \mathbb{R}^{D \times C'}$. Based on the clustering results, we can restore the corresponding relationship between the tabular attributes and the original channels of the image as $A \in \mathbb{R}^{D \times C}$. Then the channels and attributes are aligned and relevant features are selected.

## 4.3 Learning with Auxiliary Information

To leverage the knowledge of each attribute of the tabular data, we construct auxiliary tasks to learn this information. Specifically, we use the matrix $A$ to weigh the image channels, allowing us to focus the attention of the relevant tabular attributes on the corresponding image channels. We use the feature extractor of an existing image network $\phi(\cdot)$ to learn a classifier that maps from the image with a certain mask to the corresponding attributes of the tabular data. By doing so, we enhance the

image network's knowledge of the attributes of the tabular data and transfer this knowledge into the image modality. This allows the learned model to handle missing tabular modalities and improve its overall performance on complex tasks.

In summary, the loss can be written in the following form

$$
\begin{aligned}
\mathcal{L} &= \mathcal{L}(f(\boldsymbol{x}^I), y) + \mathcal{L}(g(\boldsymbol{x}^T), y) + \mathcal{L}_{i2t}, \\
\mathcal{L}_{i2t} &= \sum_p \mathcal{L}_{MSE}(\mathcal{A}_p \cdot \phi(\boldsymbol{x}^I), \boldsymbol{x}^T_{\text{num}_p}) + \sum_q \mathcal{L}_{CE}(\mathcal{A}_q \cdot \phi(\boldsymbol{x}^I), \boldsymbol{x}^T_{\text{cat}_q}).
\end{aligned} \tag{6}
$$

Here, $\mathcal{L}$ is the label prediction loss function such as cross entropy loss for classification tasks or mean square error loss for regression tasks. Since there may be numerical and categorical attributes for tabular data, we model them separately when constructing the loss to guide the image model to learn more information, expecting that the processing of different types is reasonable. The tabular model is updated in order to get a more accurate representation of each tabular attribute. $\mathcal{L}_{CE}$ is cross entropy loss for categorical attributes and $\mathcal{L}_{MSE}$ is mean square error loss for numerical attributes. This style of updating ensures that the model learns increasingly accurate channel-attribute correspondences, allowing the tabular data to guide the image data with increasing precision. By leveraging this approach, we can effectively transfer expert knowledge to images to develop more accurate and comprehensive image models for complex tasks.

To sum up, our method leverages OT to align the distributions of different modalities and select relevant tabular attributes that are closely related to the image data. We then use the alignment to enhance the image learning of the relevant attributes, thus transferring expert knowledge from the tabular data to the image model.

## 5 EXPERIMENTS

In this section, we compare CHARMS with crossmodal transfer methods on several datasets. The analysis experiment and ablations verify the effectiveness of our method. Moreover, we visualized the result of the alignment of attributes and channels.

### 5.1 EXPERIMENTS AND RESULTS

**Dataset.** Totally six datasets are used in the experiment: **Data Visual Marketing (DVM)** (Huang et al., 2022) is created from 335,562 used car advertisements. The tabular data includes some car parameters such as the number of doors and some advertising data such as the year. Different from (Hager et al., 2023), only the new version DVM dataset is available. Car models with less than 700 samples were removed, resulting in 129 target classes, a classification task. **SUNAttribute** (Patterson et al., 2014): We use the table modality in this experiment to help images more accurately predict whether a scene is an open space, which is a binary classification task. **CelebA** (Liu et al., 2015) is the abbreviation of CelebFaces Attribute, meaning celebrity face attribute dataset. It's a large-scale dataset with more than 200K celebrity images, each with 40 attribute annotations. We use Attractive as the label, which is a binary classification task. **PetFinder-adoption** dataset comes from a kaggle competition where the task is to predict the speed at which a pet is adopted, which is a five-class classification task. Tabular data contains information about the pet such as the type and vaccination status. **PetFinder-pawpularity** dataset also comes from a kaggle competition where the task was to predict the popularity of a pet based on that pet's profile and photo. **Avito** is a challenge to predict demand for an online advertisement based on its full description, its context and historical demand for similar ads in similar contexts. The target deal_probability can be any float from zero to one. It's also a regression task.

**Evaluation metrics.** For classification tasks, we compute accuracy to measure the performance. For the regression task, we use root mean square error (RMSE) for performance evaluation.

**Implementation Details.** In the course of the experiment, we implement CHRAMS with PyTorch and conduct experiments with a single GPU. Moreover, we utilize the grid search to find the hyperparameters and we choose the best models from the validation set by using early stopping. Specifically, the batch size $k$ is searched in $\{32, 64, 128\}$ and the learning rate is searched in $\{$1e-5, 5e-5, 1e-4, 5e-4, 1e-3, 5e-3$\}$. More details can be seen in Appendix A.

Table 1: Comparisons with baseline methods on DVM, SUN, CelebA, Adoption, Pawpularity, and Avito datasets. The first four are classification tasks while the last two are regression tasks. RTDL means the FT-transformer (Gorishniy et al., 2021) model trained on the tabular modality.

|  | DVM ↑ | SUN ↑ | CelebA ↑ | Adoption ↑ | Pawpularity ↓ | Avito ↓ |
|---|---|---|---|---|---|---|
| LGB | 0.9748 | 0.8501 | 0.7963 | 0.4101 | 20.0720 | 0.2290 |
| RTDL | 0.9682 | 0.8563 | 0.7936 | 0.4107 | 20.0844 | 0.2317 |
| Resnet | 0.8743 | 0.8361 | 0.8146 | 0.3477 | 18.6150 | 0.2512 |
| KD | 0.8390 | 0.8382 | 0.8118 | 0.3532 | 19.0683 | 0.2499 |
| MFH | – | 0.8312 | 0.7507 | 0.3041 | 43.1455 | 0.2873 |
| FMR | 0.8427 | 0.8347 | 0.8003 | 0.3526 | 19.3517 | 0.2937 |
| MMCL | 0.8203 | 0.8431 | 0.8041 | 0.2981 | – | – |
| CHARMS | **0.9175** | **0.8661** | **0.8220** | **0.3603** | **18.4314** | **0.2495** |

Table 2: Visualization by GradCAM. We conducted experiments on CelebA dataset and PetFinder-adoption. The results show that the OT algorithm can indeed align the tabular attributes with the image channels automatically.

| Tabular Attribute | 5_o_Clock_Shadow | Arched_Eyebrows | Big_Nose | Blond_Hair |
|---|---|---|---|---|
| Aligned Channel | 65, 87, 119, 236… | 33, 76, 78, 115, … | 50, 224, 258, … | 684 |
| Visualization | | | | |

| Tabular Attribute | Type | | Color | |
|---|---|---|---|---|
| Aligned Channel | 399, 413, 414, 521… | | 400, 412, 425, 448… | |
| Visualization | | | | |

**Results.** To demonstrate the superiority of CHARMS, we compare it with other popular methods on six datasets as shown in Table 1. The result in the form of mean plus standard deviation are shown in Appendix Table 4. Our results show that CHARMS consistently achieves the best performance on all datasets. In contrast, the baseline methods we compared with do not significantly improve the performance compared to direct training with images. In fact, some of them even decrease the results. This is likely because these methods only use the tabular data to guide the image model at a coarse level, without considering the complex relationships and interactions between the modalities. As a result, the guidance provided by these methods is not sufficient for the image model to learn useful information, which can lead to confusion and poor results.

The MFH approach only learns the KL divergence between the teacher and student networks, which may not be sufficient for handling complex tasks, as evidenced by its poor performance on the DVM 129 classification task. The experiment on the regression task is one of MMCL's limitations according to (Hager et al., 2023).

What is particularly surprising about our approach is that it can outperform the tabular modality on the SUNAttribute dataset. Similarly, on the CelebA and Pawpularity datasets, our approach can improve the performance of the image modality, even though the tabular data is weaker than images. It is possible that our approach can outperform the tabular modality even if it is a strong modality. These findings suggest that we indeed transfer tabular knowledge to images.

**Visualization.** To verify the effectiveness of OT in matching tabular attributes and image channels, we used GradCAM (Selvaraju et al., 2017) to visualize the results of OT, as shown in Table 3. On the CelebA dataset, our model can accurately capture various table attributes for the same image. On the PetFinder-adoption dataset, we demonstrate our model's ability to recognize the same attribute across different images.

Our results demonstrate that OT is able to accurately match image channels with the relevant tabular attributes, highlighting the validity of our approach in integrating tabular knowledge into the image model. This supports the rationale behind our approach and highlights the importance of carefully aligning the distributions of different modalities to effectively transfer knowledge between them.

## 5.2 Experiments Analysis

**Comparison for CHARMS and other methods.**

During the training process, we visualize the mutual information to understand how the mutual information changes during the training process. Specifically, we take ten models from the beginning of training to convergence and calculated the mutual information. The results are shown in Figure 3. Our results show that the mutual information in CHARMS increases steadily during training, demonstrating the effectiveness in transferring knowledge between modalities and improving the accuracy and interpretability of the model.

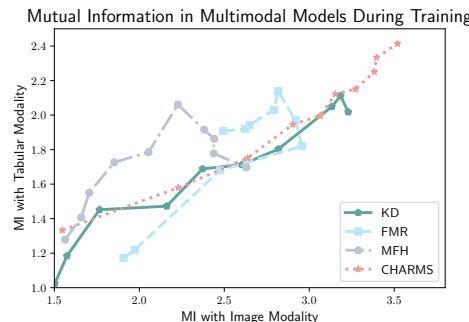

Figure 3: Mutual Information During Training on MVFEAT dataset. We calculate mutual information from the beginning to the convergence process in order to better understand the training process of each method.

Comparing our approach with the MFH and FMR methods, we found that the MFH method initially selects important features using feature importance, leading to higher mutual information with the table, but as the model focuses more on the image information, the mutual information with the table decreases. The FMR method obtains a good initialization using the tabular data, but as the table modality is down-weighted, the mutual information with both the table and image decreases.

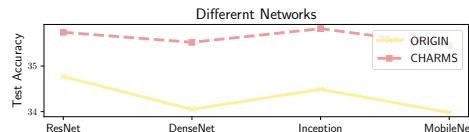

Figure 4: Impact of different network structures on the method on Adoption dataset.

Overall, visualizing mutual information provides important insights into the learning process of knowledge transfer models and can enhance the interpretability and effectiveness of these models, highlighting the importance of aligning the distributions of modalities and transferring knowledge between them. More discussions with attention method and CLIP (Radford et al., 2021) method are provided in Appendix B.

**The ablation study of components in CHARMS.** To demonstrate the applicability and robustness of our proposed method, CHARMS, we conducted experiments using different network structures, including Densenet-121, Inception-v1, and MobileNet-v2, in addition to ResNet50. Our results, shown in Figure 4, demonstrate that the performance improvements achieved by our method are consistent across different network structures, highlighting the robustness of our approach. More visualisation and interpretative experiments are provided in Appendix C.

## 6 Conclusion

In this work, we propose the CHARMS, a novel method that automatically transfers relevant tabular knowledge to images. Our method leverages tabular data as auxiliary information during transfer, enabling the transfer of expert knowledge in tabular data to images. Since not all attributes contained in tabular data are relevant to the corresponding image, we utilize optimal transport to align the attributes with channels, strengthening the correlated channels during transfer. Experimental results demonstrate that CHARMS outperforms previous methods in crossmodal transfer and our method enables insightful explanations of the learned visual embedding space with tabular instruction. We hope this work motivates future research on the challenges of multimodal encountered in real-world problems, with a particular focus on tabular data and knowledge transfer.

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

# A  EXPERIMENT DETAILS

## A.1  DATASET DETAILS

The datasets used in our experiments are MFEAT(van Breukelen et al., 1998), Data Visual Marketing (DVM)(Huang et al., 2022), SUNAttribute(Patterson et al., 2014), CelebA(Liu et al., 2015), PetFinder-adoption, PetFinder-pawpularity and Avito.

**MFEAT.** This dataset consists of features of handwritten numerals ('0'–'9') extracted from a collection of Dutch utility maps. 200 patterns per class (for a total of $2,000$ patterns) have been digitized in binary images. These digits are represented in terms of the following six feature sets. We use only 76 fourier coefficients of the character shapes and 6 morphological features for tabular data. The image modality is reconstructed from 240 pixel averages of images from $2 \times 3$ windows.

**DVM.** DVM dataset aims to facilitate business related research and applications in automotive industry such as car appearance design, consumer analytics and sales modeling. The dataset contains car images, model specifications and sales information about 899 car models that have been sold in the UK market over the last 20 years. which comprises two data parts: the image data and the table data. The former contains $1,451,784$ car images that have been deliberately cleaned and organized. While the latter includes six CSV tables that cover the non-visual attributes such as brand, price, sales, etc. Different from MMCL, only the new version DVM dataset is available(Hager et al., 2023). We pair this tabular data with a single random image from each advertisement, yielding a dataset of $70,580$ train pairs, $17,645$ validation pairs, and $88,226$ test pairs. Car models with less than 700 samples were removed, resulting in 129 target classes, classification task. There are total of 13 numerical variables and 3 categorical variables in this dataset. We expect that under the guidance of tabular data, images can learn more knowledge and make classification better.

The DVM dataset utilized in the original paper is an earlier version, and unfortunately, we don't have access to the dataset after the official update. This discrepancy in dataset versions may introduce variations in the data distribution and characteristics. Specifically, all the images are resized to 300x300 resolutions; Segment results are no longer provided directly; Image data of 2019 registered car models is added and the non-visual feature data is updated to 2020.

We follow the steps in Hager et al. (2023) in Section 4.1 to preprocess the data. In detail, the car models with less than 700 samples were removed, resulting in 129 target classes. This process ensures that the amount of data remain largely consistent with Hager et al. (2023).

Lastly, to maintain uniformity and facilitate fair comparisons, we employed a fixed batch size of 64 across all methods, whereas the original paper employed a larger 512. Additionally, we conducted MMCL method on our dataset with a batch size of 512. The result was 0.8869/0.9070. This is still somewhat different from the values reported in Hager et al. (2023) and performs worse than our method 0.9207 with a batch size of 512.

Furthermore, we conducted a comparison of GPU usage with batch size 64. Our method uses 8 GB of memory while theirs uses 20 GB. The results revealed that the MMCL method remains resource-intensive. Conversely, our method achieves superior performance with lower computational costs, further highlighting the efficiency of our approach.

**SUNAttribute.** SUNAttribute annotates 20 scenes from each of the 717 SUN categories. Each scene has 102 attributes and each attribute will have multiple annotations. For simplicity, we divide each attribute into zero and one and our goal is to predict whether a scene is an open space, which is

a binary classification task. The dataset contains $14,340$ images and the corresponding table feature, each attribute of the table feature represents a scene and takes the value of $1$ if the attribute is present in the image. we use $8:1:1$ to divide the training set, validation set, and testing set. There are total of 101 categorical variables in this dataset.

**CelebA.** CelebA is the abbreviation of CelebFaces Attribute, meaning celebrity face attribute dataset, which contains $202,599$ face images of $10,177$ celebrities, each image is well marked with features, including $40$ attribute markers such as Big_Nose. We use Attractive as the label, which is a binary classification task. We use $8:1:1$ to divide the training set, validation set, and testing set. There are total of 39 categorical variables in this dataset. We expect to introduce more detailed face information in the table, allowing the image to perform better on downstream tasks.

**PetFinder-adoption.** Animal adoption rates are strongly correlated to the metadata associated with their online profiles, such as descriptive text and photo characteristics. This dataset comes from a kaggle competition where the task is to predict the speed at which a pet is adopted, which is a five-class classification task. There are total of $10$ numerical variables and $14$ categorical variables in this dataset. Tabular data contains information about the pet such as the type and vaccination status. We also use the same division for the dataset.

**PetFinder-pawpularity.** This dataset also comes from a kaggle competition where the task was to predict the popularity of a pet based on that pet's profile and photo, which is a regression task. Each pet photo is labeled with the value of $1$ (Yes) or $0$ (No) for each of features. For example, "Face" represents whether the face of the pet in the picture is frontal. There are $12$ categorical variables in tabular data.

**Avito.** Avito, Russia's largest classified advertisements website, is deeply familiar with this problem. Sellers on their platform sometimes feel frustrated with both too little demand (indicating something is wrong with the product or the product listing) or too much demand (indicating a hot item with a good description was underpriced). This dataset is challenging you to predict demand for an online advertisement based on its full description, its context and historical demand for similar ads in similar contexts. The target deal_probability can be any float from zero to one. It's also a regression task. There are total of $2$ numerical variables such as and $11$ categorical variables such as in this dataset.

## A.2 TRAINING DETAILS

We use ResNet50 with weight pretrained on ImageNet-1k(Russakovsky et al., 2015) as image feature extractor for all methods mentioned in this paper. The classifier is built from an MLP with one hidden layer of size 1024.

For baseline methods, the numerical tabular data fields are standardized using z-score normalization with a mean value of 0 and standard deviation of 1. For our method CHARMS, we use FT-Transformer(Gorishniy et al., 2021) to get the embedding of tabular data, which can process continuous and categorical variables separately.

- **KD(Hinton et al., 2015):** For KD method, we search the temperatures in $\{1.0, 2.0, 4.0, 6.0, 8.0\}$ and $\lambda$ in $\{0.2, 0.4, 0.6, 0.8\}$.
- **KD-Fou:** This means that we use only 76 fourier coefficients of the character shapes features when training the teacher network.
- **KD-Mor:** This means that we use only 6 morphological features when training the teacher network, which can be revealed in images.
- **FMR(Yang et al., 2017):** We set ten percent of the fixed features to be knockdown in each epoch in FMR method. We search the knockdown_num in $\{0.1, 0.2, 0.3, 0.3\}$. The fixed feature classifier is a linear connection between tabular data and the corresponding image.
- **MFH(Xue et al., 2022):** For MFH method, we set modality general decisive information according to the feature ranking algorithm. The number of the features is fifty percent of that for all features.
- **MMCL(Hager et al., 2023):** The same parameters are set for MMCL method according to (Hager et al., 2023). We use the frozen version after pretrain and only train the classifier for downstream task.

Table 3: Introduction to the dataset. Here we introduce image data and tabular data in each dataset, and numerical and categorical variables are introduced separately in the tabular data. An example is given for each dataset.

| Dataset | Numerical Attribute | Categorical Attribute | Image |
|---|---|---|---|
| MFEAT | Fourier coefficient_1 0.13839 | - |  |
| DVM | Length 4865.0 | Fuel_type 9 |  |
| SUNAttribute | - | Warm 1 |  |
| CelebA | - | Big_Nose 0 |  |
| PetFinder-adoption | Fee 100 | Type 0 |  |
| PetFinder-pawpularity | - | Focus 0 |  |
| Avito | Price 1290 | Category_name 4 |  |

- **CHARMS:** For FT-Transformer, the number of Transformer blocks is set to 2. We use the K-Means method to cluster the representations obtained by ResNet50 and $n\_cluster$ is 40. Embedding dimension $E$ is set according to the data distribution. Adam optimizer with weight decay is used to train the models. We choose to update cost matrix every 5 epochs, striking a balance between updating them without stable knowledge and minimizing the computational burden. However, we continuously update $\phi$ throughout the training process to enhance the representation.

We experiment on five random seeds and the results in the form of mean plus standard deviation are shown in the Table 4.

Table 4: Comparisons with baseline methods on DVM, SUN, CelebA, Adoption, Pawpularity, and Avito datasets on five random seeds.

|  | DVM ↑ | SUN ↑ | CelebA ↑ | Adoption ↑ | Pawpularity ↓ | Avito ↓ |
|---|---|---|---|---|---|---|
| LGB | 0.9748±0.0014 | 0.8501±0.0003 | 0.7963±0.0005 | 0.4101±0.0053 | 20.0720±0.0072 | 0.2290±0.0011 |
| RTDL | 0.9682±0.0018 | 0.8563±0.0011 | 0.7936±0.0004 | 0.4107±0.0048 | 20.0844±0.0098 | 0.2317±0.0034 |
| ResNet | 0.8743±0.0183 | 0.8361±0.0144 | 0.8146±0.0092 | 0.3477±0.0048 | 18.6150±1.4559 | 0.2512±0.0034 |
| KD | 0.8390±0.0076 | 0.8382±0.0063 | 0.8118±0.0046 | 0.3532±0.0035 | 19.0683±1.7642 | 0.2499±0.0015 |
| MFH | – | 0.8312±0.0022 | 0.7507±0.0034 | 0.3401±0.0027 | 43.1455±2.0843 | 0.2873±0.0047 |
| FMR | 0.8427±0.0151 | 0.8347±0.0119 | 0.8003±0.0143 | 0.3526±0.0088 | 19.3517±1.5837 | 0.2937±0.0084 |
| MMCL | 0.8203±0.0040 | 0.8431±0.0012 | 0.8041±0.0017 | 0.2981±0.0026 | – | – |
| CHARMS | **0.9175±0.0052** | **0.8661±0.0032** | **0.8220±0.0022** | **0.3603±0.0037** | **18.4314±0.7427** | **0.2495±0.0025** |

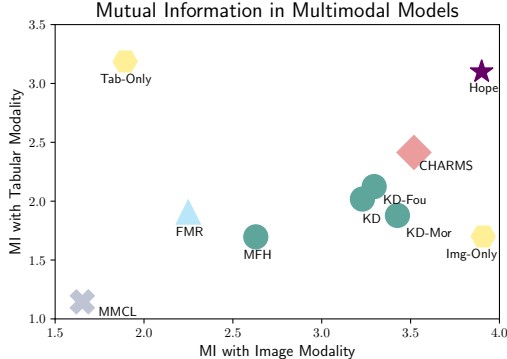

Figure 5: Mutual Information with Different Modality in Multimodal Models. A good model should be able to effectively combine both image and tabular information, resulting in higher mutual information between the two modalities.

Figure 6: Mutual Information During Training on MVFEAT dataset. We calculate mutual information from the beginning to the convergence process in order to better understand the training process of each method.

## A.3 FIGURE DETAILS

We explain some figures in detail.

- For Figure 5, we calculated the amount of information contained in different modality data for different methods with the MINE method(Belghazi et al., 2018). The image data are simple handwritten digits, we process them simply using a two-layer convolutional neural network, followed by a max pooling layer, and a Dropout layer to prevent overfitting. When calculating the mutual information, we use the $mine$ method as the loss function for approximating the mutual information. The network we choose is a three layer MLP with two hidden layers of size 100, the method we choose is $concat$, and the $batch\_size$ is 16.

- For Figure 6, we do not calculate the mutual information change process for the MMCL method because the MMCL method already performs much less well in Figure 8 than the other baseline models. We hypothesize that MMCL maps the tabular and image representations to another space and therefore the mutual information is lower.

- In the ablation study for different nets, we experimentally validated the impact of different neural network as backbone models on our approach. The accuracy in ORIGIN is {34.77, 34.05, 34.49, 33.98}. The accuracy in out CHARMS is {35.74, 35.52, 35.82, 35.45}.

## A.4 TASK DETAILS

The usage of knowledge from table to images could be explained from three aspects:

In our setting, the goal is to transfer knowledge from the tabular data to the image model. Both classification and regression tasks are vital and commonly encountered in our setting, where both of them are investigated in our experiments. For instance, on the Adoption dataset, the pet type

and size attributes are crucial for the adoption time classification. Guidance on these features in an image would lead to better learning of the image model. Similarly, on the Pawpularity dataset, the eyes and face attributes have a positive assignment on the regression of the popularity of the pet. Therefore, it makes sense to do knowledge transfer from tabular data to image for both classification and regression tasks.

CHARMS is a general method for both classification and regression tasks, in detail, we use cross entropy loss for classification task and mean square error loss for regression task. We achieved an improved image representation by employing the CHARMS method, which leverages the guidance of tabular data on the image data. Specifically, for the classification task, our approach facilitated the representation with a more discerning distribution over the target categories. On the other hand, the regression task enabled us to learn an image representation that better approximated the target values during prediction. The fact that our method performs well on both tasks underscores its generalizability and effectiveness.

Additionally, our visualization experiments provide further evidence of the effectiveness of our method. These experiments reveal that the attributes and channels selected by our approach are appropriately matched, leading to an enhancement in the performance of the image model. This alignment between the attributes and channels serves as strong evidence that we have successfully transferred the relevant knowledge from the table to the image model.

In summary, our approach demonstrates its versatility by excelling in both classification and regression tasks, showcasing its ability to enhance image representations using guidance from tabular data.

## B  ANALYSIS ON OUR CHARMS METHOD

### B.1  COMPARISON WITH ATTENTION METHOD

Our method employs the transfer matrix obtained by OT to weigh the images, with the weights of the corresponding channels raised to learn the tabular attributes. An alternative approach is to use the attention method to weigh the image channels differently and learn each tabular attribute separately, which is a more intuitive approach:

$$\phi(\boldsymbol{x}^T)_{att} = \mathcal{T}(\phi(\boldsymbol{x}^T)) \cdot \phi(\boldsymbol{x}^T) \tag{7}$$

where $\mathcal{T}$ is a two layer MLP that first downscales the image representation obtained by $\phi$ before rescaling it to its original dimension, thereby weighting the different channels of the image.

In contrast to our method CHARMS, this method assigns a weight to each input element so that the model can pay more attention to those input elements that are more important for the task at hand. The attention method constructs a learnable mask for each attribute and learns each attribute separately based on the backbone network. However, this approach may result in unequal impacts of different masks on the main task. In contrast, our method weights the attention of different channels in the representation obtained by the main task, which essentially corrects the main task while avoiding potential inconsistency issues caused by the attention method.

We compare the performance of our method CHARMS with the attention method in all experiments and summarized the results in Table 5. The table shows that the attention method did not perform as well as our method on all datasets. Specifically, on the DVM dataset, which involves a complex downstream task of 129 classification tasks, the attention method constructed different attentions for different attributes, which confused the backbone network and led to a decrease in overall task performance.

This finding highlights the impracticality of using the attention mechanism alone to integrate the abundant information in tabular data into the image model. This further supports the effectiveness of our proposed approach.

### B.2  COMPARISON WITH CLIP METHOD

CLIP is pre-trained on a large amount of text and image pairs, which makes it able to map from text to images. Some previous studies have demonstrated that CLIP is able to transform tabular

Table 5: Comparison with Attention method. Here Attention means we directly conduct the attention mechanism on the feature extracted by $\phi$ and learn an attention mask for all tabular attributes.

|  | DVM ↑ | SUN ↑ | CelebA ↑ | Adoption ↑ | Pawpularity ↓ | Avito ↓ |
|---|---|---|---|---|---|---|
| Attention | 0.4757 | 0.8550 | 0.8180 | 0.3454 | 18.7401 | 0.2544 |
| CHARMS | **0.9175** | **0.8661** | **0.8220** | **0.3603** | **18.4314** | **0.2495** |

Table 6: Comparison with CLIP method. Here CLIP-LP means two encoders are fixed, and only the classification head is trained. CLIP-FT means fine-tuning the entire CLIP network.

|  | DVM↑ | SUN ↑ | CelebA ↑ | Adoption ↑ | Pawpularity ↓ | Avito ↓ |
|---|---|---|---|---|---|---|
| CLIP-LP | 0.7619 | 0.6918 | 0.7590 | 0.3047 | 20.1537 | 0.2972 |
| CLIP-FT | 0.8417 | 0.8333 | 0.8165 | 0.2935 | 42.9489 | 0.2940 |
| CHARMS | **0.9175** | **0.8661** | **0.8220** | **0.3603** | **18.4314** | **0.2495** |

data to text for classification given the column names(Wang & Sun, 2022; Hegselmann et al., 2023). However, CLIP is heavily reliant on the semantic information contained within the text, so that the semantics of attributes are inevitable.

Indeed, the setting of this paper is more general. We expect to transfer the tabular knowledge to the image modality during training to cope with the absence of expert knowledge during testing. Our method CHARMS aims to automatically extract the semantic information from the tabular and align it with the corresponding image channels without requiring explicit knowledge of the attribute's precise meaning. Specifically, as we show in Section 4.2, based on measuring the similarity across attributes and channels, OT discovers and aligns the attribute semantic automatically.

We conducted an experiment with CLIP. In this experiment, we converted the tabular data into text format, such as "length: 16". To ensure a fair comparison, we utilized CLIP from (Radford et al., 2021) with the ResNet50 backbone. The CLIP model consists of an image encoder and a textual encoder, and we connected a one-layer linear head for classification or regression after the image encoder. Two versions of CLIP were trained in our experiment. CLIP-LP means CLIP-LinearProb, which denotes the scenario where the two encoders are fixed, and only the classification head is trained. CLIP-FT means CLIP-FineTune, on the other hand, involves fine-tuning the entire CLIP network. With the contrastive learning of the two modalities of the CLIP model, tabular knowledge is transferred to the image modality. By transforming the task into a language-to-vision knowledge transfer, the results were obtained in Table 6.

From the experiments, we can see that the performance of CLIP is not ideal. This is probably due to the fact that in tabular data, each column holds its own distinct meaning, and directly utilizing it as input to CLIP can lead to the loss of certain information. For instance, on the CelebA dataset, the attribute "wood (not part of a tree)" might not be a highly significant feature. However, when this attribute is converted to text format, its character length tends to be relatively long, which can introduce redundancy in the information.

From another perspective, previous work has pointed out that there is a modality gap in the CLIP's embedding space(Liang et al., 2022). This gap is caused by a combination of model initialization and contrastive learning optimization. In a multi-modal model with two encoders, the representations of the two modalities are clearly apart when the model is initialized. During optimization, contrastive learning keeps the different modalities separate by a certain distance. This gap makes the CLIP method fail in our task.

In summary, the loss of information and the modality gap that arises when transferring tabular data to images can hinder the performance of the CLIP method in our setting. However, our method addresses these challenges by automatically discovering and establishing the matching relationship between the two modalities, thereby facilitating effective knowledge transfer, which is a more general method.

Table 7: More Visualization by GradCAM.

| Tabular Attribute | 5_o_Clock_Shadow | Arched_Eyebrows | Big_Nose | Blond_Hair |
|---|---|---|---|---|
| Aligned Channel | 65, 87, 119, 236… | 33, 76, 78, 115, … | 50, 224, 258, … | 684 |
| Visualization | | | | |

| Tabular Attribute | High_Cheekbones | Smiling | Oval_Face | Rosy_Cheeks |
|---|---|---|---|---|
| Aligned Channel | 2, 26, 41, 85,… | 11, 12, 28, 57, … | 52, 646, 924, … | 4, 47, 88,... |
| Visualization | | | | |

| Tabular Attribute | Type | | Color | |
|---|---|---|---|---|
| Aligned Channel | 399, 413, 414, 521… | | 400, 412, 425, 448… | |
| Visualization | | | | |
| Aligned Channel | 399, 413, 414, 521… | | 400, 412, 425, 448… | |
| Visualization | | | | |

## C   MORE EXPERIMENTS

### C.1   MORE VISUALIZATION

We provide more visualizations in Table 7 to validate the ability of CHARMS to match the corresponding attributes and channels. We apply GradCAM on various datasets, which show similar visualization results, where the channels could be matched to a certain attribute with semantic meaning.

For the Adoption dataset, all tabular attributes are inherently more abstract in nature. However, for the purpose of visualization, we have specifically selected features that are visually recognizable by humans from images. For instance, attributes such as the type of pet and the color of the pet highlight more general aspects that are of interest.

From the visualization, we can see that the judgment of the pet type focuses more on the pet's head, whereas the judgment of the color takes into account the whole body of the pet, and from this point of view we believe that our approach does achieve knowledge transfer.

### C.2   VISUALIZATION WITH T-SNE

To visualize the impact of our method on the distribution of image features, we conducted experiments using the t-SNE method(Van der Maaten & Hinton, 2008). t-SNE can map high-dimensional data to a two- or three-dimensional space, enabling better visualization and interpretation of the data structure. The method employs a nonlinear mapping approach that minimizes the difference between the distances of points in high-dimensional space and those in low-dimensional space. Specifically, it represents high-dimensional data points as probability distributions and generates corresponding probability distributions in the low-dimensional space. Then, it uses KL divergence to measure the

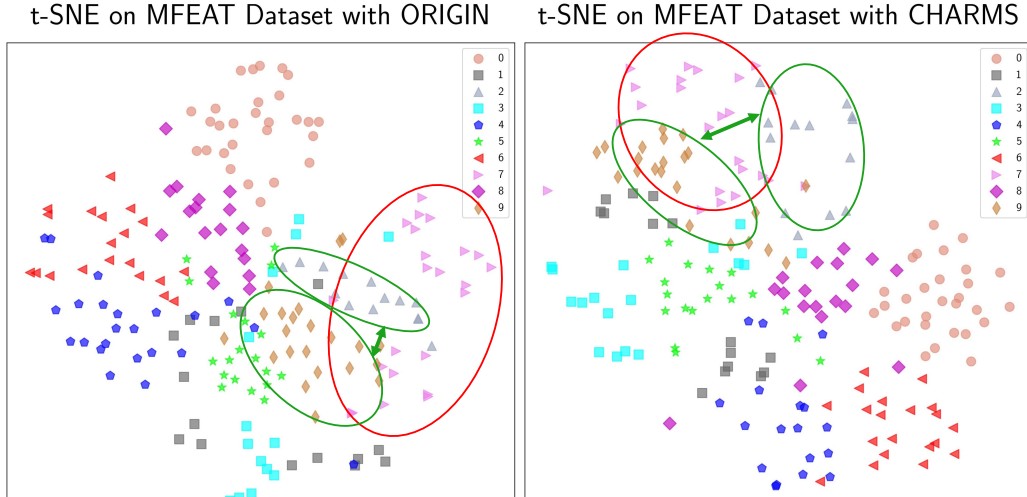

Figure 7: Visualization of t-SNE on the MFEAT dataset. the ORIGIN method represents training on image modalities only. As can be seen from the figure, our method makes the intra-class distance smaller and the inter-class distance larger. Therefore the transfer of expert knowledge from tabular data to the image model is effective. The red circles mean that our method makes the intra-class distance smaller, and the green circles indicate that our method makes the inter-class distance larger.

difference between the two probability distributions and minimizes it to achieve the best mapping effect.

The experimental results are presented in Figure 7, where the ORIGIN method refers to training with image modalities only. The figure shows that the ORIGIN method achieved good segmentation results due to the task's simplicity. However, due to the lack of expert knowledge, the intra-class distance is still large, particularly for samples with label 7, while the inter-class distances remain small, such as for samples with labels 2 and 9. In contrast, our method compensates for these deficiencies by transferring expert knowledge.

## C.3 MORE MUTUAL INFORMATION EXPERIMENTS

We chose the MFEAT dataset for the Mutual Information experiments since, in this dataset, the formal features of each category are simple and easily distinguishable. For example, morphological features and non-morphological features. And the images are all digital images, which are relatively simple and easy to understand. The experiment mainly helps us understand. More mutual information experiments can be obtained in Figure 8 9.

The experiments in PetFinder-adoption dataset also indicate that existing methods for transferring tabular knowledge to image models yield low mutual information between the representations and tabular data. Our CHARMS method, on the other hand, maximises the mutual information of tabular and images to achieve better results.

## C.4 MORE ABLATION STUDIES

In the CHARMS method, we use the K-Means(Lloyd, 1982; MacQueen, 1967) method to cluster the 2048-dimensional features extracted from ResNet. We discuss the number of clusters on the SUNAttribute dataset, and the results in Table 8 show that the performance of CHARMS is not affected by the number of clusters taken, demonstrating the robustness of the method to hyperparameter choices. This robustness makes the method more flexible and reliable in practical applications, as it does not require excessive hyperparameter tuning or fine-tuning, saving time and effort.

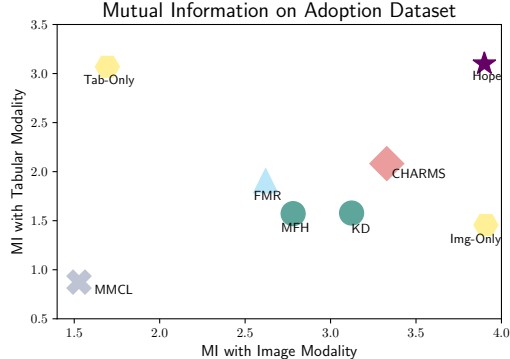 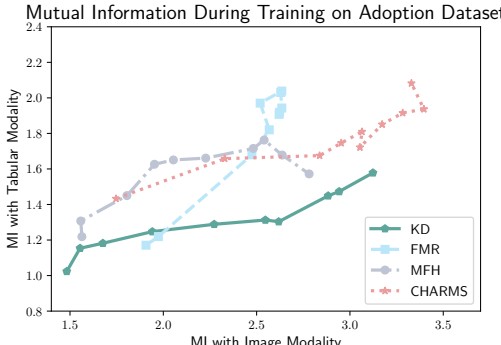

Figure 8: Mutual Information with Different Modality on the Adoption Dataset.

Figure 9: Mutual Information During Training on the Adoption dataset.

Table 8: Ablation study on cluster number on SUNAttribute dataset.

| n_cluster | 20 | 40 | 60 | 80 | 100 |
|---|---|---|---|---|---|
| Accuracy | 0.8494 | 0.8661 | 0.8494 | 0.8556 | 0.8522 |

To investigate the effectiveness of the OT method in our proposed approach, CHARMS, we conducted experiments where we reversed the transfer matrix of OT, expecting the image channels to learn the unaligned tabular attributes. We denote this approach as CHARMS-reverse. The results of this experiment are shown in Table 9, which demonstrate that the performance of CHARMS-reverse is significantly lower than that of our original method, CHARMS, highlighting the importance of OT in alignment.

To further demonstrate the applicability and robustness of our proposed method, CHARMS, we conducted experiments using different network structures on DVM dataset with results shown in Table 10. The result also shows that the performance improvements achieved by our method are consistent across different network structures.

# D  LIMITATIONS AND FUTURE WORKS

Our approach relies on leveraging mutual information between the two modalities, which establishes the feasibility of knowledge transfer. When there is a significant amount of mutual information present between the tabular and image modalities, our approach can effectively transfer relevant knowledge and insights between them. On the other hand, converting text into tables is indeed a viable approach, but this approach results in the loss of some of the textual information and it is challenging to handle such a conversion well. The problem of testing data drift also exists in real life. We will consider this problem deeply in future work. In terms of social impact, we think that our approach holds potential for application in the medical field, where it can assist doctors in making rapid and accurate diagnoses. There should be no negative social impact of our method.

Table 9: Ablation study on Optimal Transport. CHARMS-reverse means that we reverse the transfer matrix of OT and make channels and attributes misaligned. The performance degradation proves that alignment is important.

| | DVM ↑ | SUN ↑ | CelebA ↑ | Adoption ↑ | Pawpularity ↓ | Avito ↓ |
|---|---|---|---|---|---|---|
| CHARMS | 0.9175 | 0.8661 | 0.8220 | 0.3603 | 18.4314 | 0.2495 |
| CHARMS-reverse | 0.8865 | 0.8459 | 0.8165 | 0.3440 | 18.8068 | 0.2568 |

Table 10: Impact of different network structures on the method on DVM dataset.

|                | ResNet | DenseNet | Inception | MobileNet |
|----------------|--------|----------|-----------|-----------|
| Model Size / M | 25.8   | 8.2      | 6.8       | 3.7       |
| ORIGIN         | 0.8743 | 0.8671   | 0.7492    | 0.8206    |
| CHARMS         | 0.9175 | 0.9115   | 0.9012    | 0.8961    |

Our work demonstrates the effectiveness of our method in both classification and regression tasks. In future work, it would be valuable to investigate the applicability of our method to other tasks, such as semantic segmentation. These types of tasks may require additional domain-specific knowledge, such as precise object localization within images, to achieve optimal performance. Nonetheless, we believe that our approach is still applicable for such tasks.

On the other hand, the high cost of annotating expert data often leads to imbalanced datasets, which pose a challenge for improving image model performance using a limited amount of tabular data. Therefore, addressing this data imbalance is crucial for future work.

