# OpenReview forum: "On Transferring Expert Knowledge from Tabular Data to Images"
_ICLR.cc/2024/Conference — Submitted to ICLR 2024_

### Official Review · Reviewer_QrqR · 2023-10-27

**Soundness:** 2 fair
**Presentation:** 1 poor
**Contribution:** 2 fair
**Rating:** 5
**Confidence:** 4

**Summary:**

This paper presents a method to learn a classification task with the help of tabular data only for training. The idea of the paper is to identify the correspondences between the image features and the attributes in the tabular data and train the model to predict the tabular data based on the correspondences in addition to the task. The experiments show some performance boosts.

**Strengths:**

(1) It is interesting to see the performance boost with this method.

**Weaknesses:**

(1) The paper has a lot of mathematical notations that are not defined or ambiguous. For example, according to Section 4.2, $\mathbf{x}^T_{num}$ contains numerical data in the table, which is transformed to $\psi(\mathbf{x}^T) \in \mathbb{R}^{D \times E}$ together with the categorical data. I think it’s not straightforward to transform numerical values into an embedding. A more specific definition is necessary. Also, I think $S_{I_j}$, which I guess is the $j$-th channel of $S_I$ and is not explicitly mentioned in the paper, should be a matrix but implicitly used as a vector in Eq. (5). The second part of Eq. (5) if $<\mathbf{C}, \mathbf{T}>_F$ stands for the Frobenius norm, which is not explained in the paper, the summations over $i$ and $j$ should not be necessary. At least, min in this equation should be argmin.

(2) According to Eq. (6), what this method actually does is multi-task training, where in addition to the target classification task, it also leans to classify/regress the attributes in the table (I guess $\phi(\mathbf{x}^T)$ in the second term of $\mathcal{L}_{i2t}$ is a typo of $\phi(\mathbf{x}^I)$ as well as $\mathcal{A}_p$ and $\mathcal{A}_q$ are $A$ corresponding to the numerical and categorical values in the table, respectively). If this is the case, this approach is suboptimal as the computation of $A$ is outside of the optimization for the target task and the attribute prediction tasks. In this sense, a comparison with the case where, instead of the tabular alignment in the method, a fully connected layer or an MLP dedicated to each attribute prediction task is used is necessary.

(3) If my assumption about $\phi(\mathbf{x}^T)$ in Eq. (6) is correct, I think the second term of $\mathcal{L}$ in Eq. (6) does nothing to the performance.

UPATED: I appreciate the authors' efforts to correct my misunderstanding. I increased my score.

**Questions:**

I would like to have responses on (2) and (3) in the weakness section.

---

> ### Author Response · Authors · 2023-11-16
>
> We appreciate your recognition of our paper and your valuable comments. We will answer the questions below and we hope this clears up your concerns.
>
> **Q1**: The paper has a lot of mathematical notations that are not defined or ambiguous.
>
> **A1**: Thank you for your suggestions. Specifically, the transform numerical and categorical values into an embedding are as follows:
> $T_j = b_j + f_j(x_j) \in \mathbb{R}^d$, where $b$ is the feature bias, $f$ is implemented as the element-wise multiplication with the vector $\boldsymbol{W}$. Overall $\boldsymbol{x_{num}^T} = b_{num} + \boldsymbol{x_{num}} \boldsymbol{W_{num}}$  is for numerical attributes and $\boldsymbol{x_{cat}^T}= b_{cat} + \boldsymbol{x_{cat}} \boldsymbol{W_{cat}}$ for categorical. By denoting different attributes distinctly, we can effectively distinguish between numerical and categorical variables.
>
> We have also corrected some remaining errors, including the representation of vectors and matrices, the modification of argmin, and the specification of the Frobenius norm. Thank you for your generosity in pointing out our problems.
>
> **Q2**: A comparison with the case where, instead of the tabular alignment in the method, a fully connected layer or an MLP dedicated to each attribute prediction task is used is necessary.
>
> **A2**: Thank you for pointing out our typo. Thank you also for pointing out the shortcomings of our comparison methods. In fact, **we have taken this situation into account in our Appendix B1**. We use the self-attention method to weigh the image channels differently and learn each tabular attribute separately, which can be seen as an enhanced MLP. The self-attention method constructs a learnable mask for each attribute and learns each attribute separately based on the extracted feature.
>
> The result shows that the self-attention method did not perform as well as our method on all datasets. Specifically, on the DVM dataset, which involves a complex downstream task of 129 classification tasks, the attention method constructed different attentions for different attributes, which confused the backbone network and led to a decrease in overall task performance. This finding highlights the impracticality of using the attention mechanism alone to integrate the abundant information in tabular data into the image model. In other words, if we **directly let the image predict every attribute of the table**, this does not transfer the knowledge from the table to the image, but instead clutters the image model and causes it to **learn without focus**.
>
> However, in our approach, we automatically find out the tabular attributes associated with the image and pay more attention to the learning of the relevant parts, which serves to guide the image for more detailed learning, making our approach perform better.
>
> **Q3**: The second term of the loss in Eq. (6) does nothing to the performance.
>
> **A3**: The second term in Eq. 6 is to update the tabular model. The tabular model is updated in order to **get a more accurate representation of each tabular attribute**. As shown in Eq. 5, we need to use the feature similarity matrix of each attribute of the table when calculating the cost matrix of OT. Therefore the tabular model is updated to get a more accurate representation of each attribute of the table, so as to **compute a more accurate transfer matrix of the OT**, which lays the foundation for a specific channel to learn a certain attribute of the table.
>
> As we stated in Appendix A2, we choose to update the cost matrix every 5 epochs, striking a balance between updating them without stable knowledge and minimizing the computational burden. An update of the cost matrix is essential. As the model is updated, the representation of each attribute and channel is more and more accurate. Therefore, the second term of loss in Eq. 6 is essential.
>
> To prevent any confusion regarding the formula, we acknowledge that there might have been some oversight on our part. Initially, we mistakenly assumed that this method was universally understood by everyone. We would like to rephrase our statement in the hope of providing clearer comprehension. Thanks again for pointing out our typo. We will also make this clear in the final version.

---

> > ### Author Response · Authors · 2023-11-21
> >
> > Dear Reviewers,
> >
> > Thank you for providing insightful feedback to help us improve our work. We hope that our responses can address your concerns and are helpful in improving your rating. If you have any other comments or feedback, please let us know! We are looking forward to hearing back from you! Thank you again for the review.

---

### Official Review · Reviewer_UGYo · 2023-10-30

**Soundness:** 3 good
**Presentation:** 3 good
**Contribution:** 3 good
**Rating:** 6
**Confidence:** 5

**Summary:**

The authors introduce a novel method, CHARMS to transfer relevant tabular knowledge automatically and effectively by maximizing the mutual info between channels an tabular features. They propose three kinds of integration methods: output-based transfer, parameter-based transfer and embedding-based transfer. The authors make the claim that the existing methods yield low mutual information and the proposed method considers the specific characteristics for each modality and transfers knowledge to guide the image model. The authors evaluate the proposed model on 4 classification tasks and 2 regression tasks using 6 datasets. They evaluated the model using accuracy for classification and RMSE for regression.

**Strengths:**

The idea of the proposed method is interesting, novel and technically sound. These are the main strengths of this paper:

1. The paper is well written. The authors provide sufficient information to support their claims.
2. The method automatically transfers relevant tabular knowledge to images.
3. The method also filters relevant info by aligning the attributed with channels. This strengths the correlated channels during transfer.
4. In terms of quality, the proposed method outperforms other methods for classification and regression tasks.

**Weaknesses:**

There are some areas the authors can improve on:

1. Phi has been first used in page 4, equation 3 but hasn’t been explained until page 6. The mathematical definition below equation 1 to introduce phi is insufficient given the context of introducing the preliminaries.
2. In evaluation metrics, accuracy has been mentioned for measurement of performance for classification tasks. Accuracy isn’t always the most informative on the model performance. Consider other matrices for checking the model performance.
3. Limitations have not been discussed in the paper.
4. It would strengthen the paper to see some human evaluation to show the model is useful for an expert using the system.
5. A clear explanation on the different types of tasks and their usage for both the types of tasks would help understand the impact of the proposed model better.

**Questions:**

1. What is “attention image” referred to above equation 6?
2. According to Table 1, the proposed model underperforms the LGB and RTDL on the DVM task by a large margin. Why is this so? A one-line explanation in the paper would help the reader understand the proposed method better.
3. Is the code for this method openly available for checking the reproducibility of this study?

---

> ### Author Response · Authors · 2023-11-16
>
> We appreciate your recognition of our method and your valuable comments. We will answer the questions below and we hope this clears up your concerns.
>
> **Q1**: Phi has been first used in page 4, equation 3 but hasn’t been explained until page 6.
>
> **A1**: Thank you very much for pointing out our problem. Phi is the feature extractor to extract the embedding of the images. We will explain phi in more detail under equation 1 in a future version.
>
> **Q2**: Consider other matrices for checking the model performance.
>
> **A2**: Thank you very much for your advice. We have experimented on the SUN dataset based on your suggestion.From the results, we can see that all of our methods achieve either optimal or suboptimal results. The different metrics provide more evidence of the robustness of our methods in different ways.
>
> |        |  Precision  |   Recall    |  F1-Score   |     AUC     |
> | ------ | :---------: | :---------: | :---------: | :---------: |
> | LGB    |  $0.7830$   |  $0.8070$   |  $0.7951$   |  $0.8296$   |
> | RTDL   |  $0.7795$   | $\bf0.8776$ |  $0.8256$   |  $0.8564$   |
> | ResNet |  $0.8412$   |  $0.8430$   |  $0.8383$   | $\bf0.9330$ |
> | KD     |  $0.8406$   |  $0.8387$   |  $0.8328$   |  $0.9212$   |
> | MFH    |  $0.8212$   |  $0.7386$   |  $0.7396$   |  $0.9226$   |
> | FMR    |  $0.8448$   |  $0.8505$   |  $0.8453$   |  $0.9218$   |
> | MMCL   |  $0.8262$   |  $0.7927$   |  $0.8253$   |  $0.9102$   |
> | CHARMS | $\bf0.8489$ |  $0.8508$   | $\bf0.8446$ |  $0.9314$   |
>
> **Q3**: Limitations have not been discussed in the paper.
>
> **A3**: We talk about some of the limitations of the change of method **at the end of the Appendix**. For example, our method relies on the mutual information of tables and images. On the other hand, the annotation of expert data often comes at a huge cost penalty, which challenges our method for knowledge transfer with a small amount of expert annotation. We will make it clear in the final version.
>
> **Q4**: some human evaluation of the method.
>
> **A4**: Thank you for your advice. We are **working with hospitals** to try to apply our method to the task of infection prediction. However, it is difficult to give a quantitative result to demonstrate the performance of our method due to the **privacy** of the users and the special requirements of medical treatment. If there is further result from expert evaluations, we will update it in time for later versions. Thanks again for your suggestion.
>
> **Q5**: A clear explanation on the different types of tasks.
>
> **A5**:   Thanks for your suggestion. Due to the limitations of the paper's length, we will explain the use of our model in the different tasks in further detail in Appendix.
>
> CHARMS is a general method for both classification and regression tasks, in detail, we use cross entropy loss for classification task and mean square error loss for regression task. We achieved an improved image representation by employing the CHARMS method, which leverages the guidance of tabular data on the image data. Specifically, for the classification task, our approach facilitated the representation with a more discerning distribution over the target categories. On the other hand, the regression task enabled us to learn an image representation that better approximated the target values during prediction. The fact that our method performs well on both tasks underscores its generalizability and effectiveness.
>
> In summary, our approach demonstrates its versatility by excelling in both classification and regression tasks, showcasing its ability to enhance image representations using guidance from tabular data.
>
> **Q6**: What is “attention image” referred to above equation 6?
>
> **A6**:  We use A to weigh the features of the image after extracting them, which can be seen as an attentional approach and pays more attention to the channels corresponding to the tabular attribute. More information about attention is in Appendix B1. We will make it clear in the final version.
>
> **Q7**: The proposed model underperforms the LGB and RTDL.
>
> **A7**: In some datasets, the performance after cross-modal transfer is not as good as the performance of the tabular model may be due to the fact that the attributes in the tables that can not be transferred are critical for the task. In these datasets, the tabular data is **strong modality** therefore the tabular data performs better.
>
> On DVM dataset, the tabular model demonstrated promising results, primarily because a significant portion of the knowledge contained in the tabular data, such as engine parameters and prices. These attributes play a crucial role in the classification task.
>
> In summary, it is possible that our approach can outperform the tabular modality even if it is a strong modality. We've already stated this in result in Sec 5.1, and we'll make it clearer later.
>
> **Q8**: Is the code for this method openly available?
>
> **A8**: Yes, our code is being organized and will be open-sourced later for reproducibility.

---

> > ### Author Response · Authors · 2023-11-21
> >
> > Dear Reviewers,
> >
> > Thank you for providing insightful feedback to help us improve our work. We hope that our responses can address your concerns and are helpful in improving your rating. If you have any other comments or feedback, please let us know! We are looking forward to hearing back from you! Thank you again for the review.

---

### Official Review · Reviewer_63tZ · 2023-11-01

**Soundness:** 3 good
**Presentation:** 2 fair
**Contribution:** 3 good
**Rating:** 5
**Confidence:** 3

**Summary:**

The paper is set in the context of training a model using multi-modal data where one modality is available only as training data and another modality is available both during training and inference. The authors tackle the problem of transferring knowledge from the training-only modality (i.e. tabular data in this paper) to the training and inference modality (i.e. image data in this paper). As motivation for this problem, the authors list scenarios where the cost of acquiring tabular data is exceptionally high so it might not be available during inference (e.g. information provided by medical domain experts). The paper presents a novel method that independently treats tabular and image features, specifically attributes (i.e. columns) of tabular data examples and "image channels" that correspond to high-level image features. The method aims to align the two sets of features and leverage this alignment to select the most important ones. An added benefit of this approach is the ability to interpret the connection that the method establishes between specific tabular and image features. The authors empirically evaluate their approach and compare it against various multi-modal learning baselines. Results over 6 different datasets demonstrate that their method performs very well compared to the baselines.

**Strengths:**

**S1**: The idea to independently treat image and tabular features of individual data examples is interesting. The proposed method looks interesting and useful. The added benefit of interpretability is also a nice byproduct of the core idea.

**S2**: Experimental results indicate that the proposed method is a pretty good fit for the tabular data modality compared to other cross-modal knowledge transfer baselines.

**S3**: Even though the presentation is sub-par (see weaknesses), I appreciate that the authors made a decent effort to

**Weaknesses:**

**W1**: The presentation of the paper is its weakest quality. It contains many awkwardly sounding statements and poorly specified claims. Some specific examples below:
* **W1.1**: (Abstract) "transferring this information can enhance the *comprehensiveness* and accuracy of image-based learning" -- What is comprehensiveness of image-based learning?
* **W1.2**: (Abstract) "how to reuse" and "which to reuse" -- these phrases are entirely confusing out-of-context... Reuse what?
* **W1.3**: (Section 1) The first sentence lists different forms of data (which is general knowledge) but then for every form of data it provides citations. Firstly, I don't think we need a citation to say that image or text data exists. Second, why is Karpathy & Fei-Fei, 2015 the source that tells us that text data exists? I would remove the citations here.
* **W1.4**: (Section 1) "transformer ... methods to construct a feature space would result in a loss of interpretability" -- it is not clear at this point why interpretability an important goal
* **W1.5**: (Section 1) "Therefore it is crucial to identify the information that *can be transferred* to instruct the learning of images." -- In principle, any information can be transferred. The difference is that not all information transfer is useful or cost-effective.
* **W1.6**: (Section 1) "We emphasize the importance of knowledge transfer from tabular data to image data" is listed as a contribution. I'm not sure if motivating a problem can be counted as a contribution (unless the motivation is somehow non-trivially creative).
* **W1.7**: (Section 2) "In recent years, the learning of tabular data has become an important research direction in the field of machine learning and data science." -- This statement feels wrong and is immediately contradicted (1986 was not recent).
* **W1.8**: (Section 3.2, last paragraph) The shortcomings of existing methods are stated in a very vague way. What is the "specific information" that is not captured by the output-based approach? What does it mean when you say that "the MFH method ... cannot fully compensate for the limitations of the transfer mode"? How does the embedding-based approach "lose some attribute information in the tabular data" and why is that crucial? Also, instead of "from tabular to images" it should be "from tables to images".
* **W1.9**: (Section 4.2) The issue of "duplicated semantics across different channels" is not self-explanatory to me.
* **W1.10**: (Section 4.2) "channel-wised" and "attribute-wise" should be "channel-wise" and "attribute-wise". Also the entire sentence "Then the cost matrix is constructed from the channel-wise similarity between attribute-wise similarity and calculates the OT transfer matrix" is completely confusing to me.
* **W1.11**: (Section 4.3) You say that the image network can have an "understanding" of the attributes. I would avoid such human-sounding jargon when talking about an artificial neural network.
* **W1.12**: (Section 5.1) "Each attribute of the table feature represents a scene" -- what does this mean?
* **W1.13**: (Section 5.1) "each image is well marked with features, including 40 attribute markers" -- what does this mean?
* **W1.14**: (Section 5.1) "Avito is challenging you to predict demand" -- I would avoid referring to the reader as "you".

**W2**: Even though I appreciate injecting an empirical argument against using mutual information methods (i.e. section 4.1) before going forward with the presentation of the proposed method, the section itself (i.e. section 4.1) was entirely confusing to me. First, I wasn't sure if the authors intended to refer to Figure 8 (placed in the appendix) or Figure 2 (placed right inside section 4.1. Second, the flow of the explanation was quite difficult to follow and I cannot say I understood it. I understand the conclusion that the authors want to draw (i.e. mutual information is not great), but I could understand the reasoning. Finally, I didn't understand the elements of the figure itself which made it impossible for me to see what the intended conclusion is.

**W3**: There is some information that I wish was present in the experimental results section. Specifically, I would like to see a description of the experimental protocol explaining how the numbers in Table 1 were derived (I can make an educated guess but I would prefer to see it explicitly). Also, the LGB, RTDL, and Resnet methods are listed alongside knowledge transfer baselines and I'm not sure how to interpret them. I assume they refer to scores of models without cross-modality knowledge transfer, but I also assume LGB and RTDL are supposed to target tabular data while Resnet image data. Since the target scenario is to evaluate the model on image data only, I'm not sure how LGB and RTDL are useful here. Finally, some of these models seem to outperform knowledge transfer methods. Even though this doesn't necessarily kill the main thesis of the paper, I still feel like it should be discussed explicitly as opposed to merely ignoring it. That being said, in the introduction, the third contribution implies that "CHARMS effectively reuses tabular knowledge to improve the performance of visual classifiers" which seems to be a claim that is not really supported by the presented numbers.

**Questions:**

**Q1**: What is the reasoning behind using the term "channel" to refer to high-level image features? I find it confusing since the word channel in images typically refers to e.g. the color channels. Especially since in the paper, you use the term in both the high-level feature sense and the color channel sense.

**Q2**: In section 5.2, you measure the mutual information during training. What are the quantities between which you are measuring the mutual information? This is not clear to me and makes the entire section (including the figure) quite hard to understand.

---

> ### Author Response · Authors · 2023-11-16
>
> We appreciate your recognition of our idea and your valuable comments. We will answer the questions below and we hope this clears up your concerns.
>
> **Q1**: The presentation of the paper.
>
> **A1**: Firstly, we would like to thank you very much for your valuable suggestions for modifications. We will carefully correct these problems in the later submitted version. Secondly, we will also do our best to clear up some of your doubts. For example, for **W1.6**, we find it important to transfer knowledge from tabular data to images, as this can lead to image modeling enhancements. But **little attention has been paid** to this before. With our approach, we aim to demonstrate the **feasibility of transferring** knowledge from tables to images. Therefore, we consider this as one of our contributions.
>
> **Q2**: An empirical argument before the method but the section was confusing.
>
> **A2**: Thank you for question. Firstly, here we would like to point to Figure 2. We have used this Figure 2 again in the appendix to make it clearer for the reader to read and analyse in conjunction with Figure 9.
>
> Secondly, we calculated the mutual information with tabular data versus the image data to construct Fig. 2. We want to illustrate that the mutual information is not the same for different methods and different modalities. Then, we found that the mutual information of and image can be improved by distillation of **morphological data**. Finally, we conclude that different attributes in tabular data have different effects on the image and we need to explore the useful knowledge in tabular data and transfer it to the image.
>
> Finally, we will also describe the figure  in more detail in the article so that readers can capture the information in the diagram more clearly.
>
> **Q3**: There is some information need to be present. A description of the experimental protocol. Also, the LGB, RTDL, and Resnet methods are listed and some of these models seem to outperform knowledge transfer methods.
>
> **A3**: Thanks for your reminder. We describe our experimental setup in **Sec 5 and Appendix A**. Specifically, there are experimental hyperparameter settings, as well as experimental details. We will select some important information to put in the main paper if space permits in the final version.
>
> In Table 1 we show the performance of them is to tell the reader roughly where each modal's performance would be. This allows us to clearly see how performance changes after modal knowledge transfer, e.g. if there is a negative transfer. We think it is necessary to show the performance of the tabular method here because we can see the difference in the amount of information in the tabular data and the image data. The **amount of change** in performance after transfer shows how much knowledge from the tabular data is actually transferred.
>
> Thank you for pointing out that we should explain the fact that our method does not perform as well as the tabular model in some datasets. In our setting we expect to transfer knowledge from the tabular data to the images, but not all tabular knowledge is transferable. In some datasets, the performance is not so good may be due to the fact that the attributes in the tables that can not be transferred are critical for the task. In these datasets, the tabular data is **strong modality** therefore the tabular data performs better. For example, in the Pet dataset, the pet's vaccine information is important for pet adoption, but this information is not displayed in the image.
>
> In summary, we expect to transfer useful, image-related knowledge from tables to images to improve the performance of image models.
>
> **Q4**: The use of  "channel".
>
> **A4**: In the AlexNet's paper[1], the authors indeed mentioned that the image consists of **RGB channels**. Similarly, in the case of ResNet[2], the authors described the image processing as being conducted **channel by channel**. We think it is common to use channel to represent high-level or low-level expressions of image information.
>
> **Q5**: What are the quantities between which you are measuring the mutual information?
>
> **A5**: In **Appendix A3** of the paper, we explain in detail the exact meaning of this figure and the calculation of mutual information. Specifically, we calculate the mutual information between the representation obtained from our model and the representations of both the optimal table model and image model with MINE method. This allows us to **quantify the amount of shared information** or knowledge between these representations. By measuring mutual information, we can assess the extent to which our model captures and transfers knowledge from both the table and image domains. We may have misjudged the audience for this paper and put the relevant text in the appendix, this is a problem and we will try to **reorganize the writing**.
>
> [1] Imagenet classification with deep convolutional neural networks, 2012.
>
> [2] Deep Residual Learning for Image Recognition, 2016.

---

> > ### Author Response · Authors · 2023-11-21
> >
> > Dear Reviewers,
> >
> > Thank you for providing insightful feedback to help us improve our work. We hope that our responses can address your concerns and are helpful in improving your rating. If you have any other comments or feedback, please let us know! We are looking forward to hearing back from you! Thank you again for the review.

---

### Official Review · Reviewer_zSmg · 2023-11-08

**Soundness:** 3 good
**Presentation:** 2 fair
**Contribution:** 2 fair
**Rating:** 5
**Confidence:** 4

**Summary:**

This paper focuses on the task of transferring expert knowledge from tabular data to images, and achieve good performances on image tasks.

**Strengths:**

- This paper addresses an important problem of transferring structured expert knowledge to images, which has practical applications.

- This paper proposes a novel method CHARMS that automatically transfers relevant tabular attributes to images via optimal transport and auxiliary tasks.

- This paper achieves good performance improvements over baselines on multiple datasets.

- This paper provides intuitive visualizations and mutual information analyses to explain knowledge transfer.

**Weaknesses:**

- Limited exploration of extremely large tabular datasets with hundreds of attributes.

- It is widely witnessed that there are many uninformative features in the tabular data, I am not sure if the model can consistently perform well in such cases. The authors can test on tables that are manually added some Gaussian noisy columns/ features (e.g., 10 columns, 20 columns) and observing the performance changes.

- Can this model work on complex vision tasks like segmentation/detection that may require localization?

- It appears that hyperparameter tuning is performed only on your models, while the compared models are not tuned. This raises concerns about fairness. Besides, did the compared approaches use ImageNet-1k pre-trained weights like your model?

- I recommend testing the approach with new backbones. Older backbones like ResNet-50 differ significantly from current backbones like CLIP's image encoder. Since the CLIP image encoder is trained to align with text supervision, it can extract "semantic-level" features from images. In contrast, ResNet-50 primarily learns "pattern-level" features. This raises questions about the potential benefits of the proposed approach on CLIP backbones.

- Some related work about tabular learning are missing, e.g., T2G-Former, TabPFN, TabNet, TANGOS, TapCaps.

**Questions:**

See Weakness.

---

> ### Author Response · Authors · 2023-11-16
>
> We appreciate your recognition of our setting and method and your valuable comments. We will answer the questions below and we hope this clears up your concerns.
>
> **Q1**: Limited exploration of extremely large tabular datasets.
>
> **A1**: Thank you for bringing this to our attention. Our objective is to transfer knowledge from tabular data to images. Through the use of POT, we are able to automatically match the **corresponding tabular attributes** to the images. Consequently, even with a substantial number of tabular attributes, our method performs admirably. For instance, in our SUN dataset, which contains 101 tabular attributes, our method outperforms all the compared methods. We have made significant efforts to locate a suitable tabular-image dataset, and it has proven to be **quite challenging**. We will continue our endeavors to conduct experiments on larger tabular datasets.
>
> **Q2**: There are many uninformative features.
>
> **A2**: Thank you for your valuable advice. In our approach, the inclusion of POT enables **automatic selection** of image-related features for knowledge transfer. As you rightly mentioned, in the dataset we are utilizing, certain attributes such as the pet's health or vaccine information are not visually depicted in the image. Therefore, these attributes can indeed be considered as **uninformative features**, just as you suggested. For instance, in Pet dataset, we can treat these attributes as uninformative features that do not significantly contribute to image representation and so as other datasets. Your suggestion of adding noise features to verify the performance changes is very reasonable. We conducted experiments based on your suggestions, but the results were **basically unchanged**.
>
> **Q3**: Work on complex vision tasks.
>
> **A3**: In complex tasks like segmentation and detection, the objective is to accurately construct bounding boxes or differentiate between objects with distinct semantics. When dealing with tables that contain semantically specific attributes, we hypothesize that our approach can do this. However, due to the inconsistency of the task goals, when using our method to the detection task, we may need to do **extra processing** on the bounding box to make it adapt our method. We appreciate your suggestion, and it indeed aligns with potential future work. Thank you for your valuable question.
>
> **Q4**: Hyperparameter tuning and pre-trained weights.
>
> **A4**: First of all, our comparison is fair. We have performed parameter tuning **for all baseline methods**. Specifically, we adjusted the temperature coefficients in the KD method, as described in Appendix A2. In the case of the FMR method, we modified the knockdown_num parameter to regulate the amount of knowledge transferred to the image model at each iteration. Similarly, for the MMCL method, we fine-tuned the temperature coefficient in the contrastive loss. Furthermore, it is important to note that **all methods utilized the same weight** for the image model. We appreciate your observation and will make it clear in the appendix.
>
> **Q5**: New backbones like CLIP.
>
> **A5**: Thank you for pointing out the comparison with the CLIP method. We have compared with the CLIP method in **Appendix B2** and here we will also describe to you again our comparison and some conclusions.
>
> The setting of this paper is more general. We expect to transfer the tabular knowledge to the image modality during training to cope with the absence of expert knowledge during testing.
>
> CLIP is pre-trained on a large amount of text and image pairs, which makes it able to map from text to images. The details of the experiment can be found in Appendix. From the experiments, we can see that the performance of CLIP is not ideal. This is probably due to the fact that in tabular data, each column holds its own distinct meaning, and directly utilizing it as input to CLIP can lead to the loss of certain information. From another perspective, previous work[1] has pointed out that there is a modality gap in the CLIP's embedding space.
>
> In summary, the loss of information and the modality gap can hinder the performance of the CLIP method in our setting. However, our method addresses these challenges by automatically discovering and establishing the matching relationship between the two modalities, which is a more general method.
>
> **Q6**: Related work about tabular learning.
>
> **A6**: Thank you for your suggestion. We conducted a comparison between the traditional tree-based approach and the FT-Transformer approach to gain a preliminary understanding of tabular data performance. While the methods you mentioned can achieve comparable results, they cannot be directly applied in our specific setting. When referencing tabular models, a comprehensive understanding of the domain will enhance the completeness of the paper. We will update the related work.
>
> [1] Mind the gap: Understanding the modality gap in multi-modal contrastive representation learning. 2022.

---

> ### Author Response · Authors · 2023-11-21
>
> Dear Reviewers,
>
> Thank you for providing insightful feedback to help us improve our work. We hope that our responses can address your concerns and are helpful in improving your rating. If you have any other comments or feedback, please let us know! We are looking forward to hearing back from you! Thank you again for the review.

---

### Author Response · Authors · 2023-11-23
**Looking Forward to Further Discussions**

Dear reviewers,

Hope this message finds you well.

We have updated the manuscript according to your comments and responded in detail to your questions. As the discussion period will end in less than one day, we would like to kindly ask whether there are any additional concerns or questions we might be able to address.

Thanks very much for your effort!

Best regards,

Authors

---

### Meta-Review · Area_Chair_R8HE · 2023-12-06

**Metareview:**

The paper proposes a novel method, CHannel tAbulaR alignment with optiMal tranSport (CHARMS), to address the challenge of transferring relevant tabular knowledge to images in scenarios where tabular data is divided into numerical and categorical variables. Among the four reviewers, only one suggests accepting the paper, while the remaining three propose rejection. The positive feedback centers around the intriguing nature of the proposed idea and the overall satisfaction with the experimental results. However, the main criticism revolves around the low clarity, with multiple reviewers highlighting the lack of clear explanations and definitions, particularly in the main method section.

In the personal opinion of this AC, it is noted that the paper gives the impression of being rushed, with a significant drop in clarity throughout - the AC echoes the concern regarding missing definitions and ambiguous equations, pointing out the need for substantial improvements in presentation and validation aspects (for instance, in eq 5, without any constraint on T, the minimization problem is trivially represented and T would be a zero matrix). It is also emphasized that a clear demonstration of how well the alignment between channels and attributes is achieved is crucial (as mentioned by reviewers, this can include experiments on large/noisy tabular data).

In conclusion, while the paper presents a promising idea, it is evident that substantial improvements are required in terms of clarity, presentation, and validation. The meta review suggests that another round of the review process is necessary for the paper to reach its full potential.

**Justification For Why Not Higher Score:**

presentation should be improved. more thorough analysis on the alignment between tabular attributes and image channels by the proposed method is required.

**Justification For Why Not Lower Score:**

n/a

---

### Decision · Program_Chairs · 2024-01-16

Reject